# Emergency Craniotomy and Burr-Hole Trephination in a Low-Resource Setting: Capacity Building at a Regional Hospital in Cambodia

**DOI:** 10.3390/ijerph19116471

**Published:** 2022-05-26

**Authors:** Jingjing Hu, Vannara Sokh, Sophy Nguon, Yang Van Heng, Hans Husum, Roar Kloster, Jon Øyvind Odland, Shanshan Xu

**Affiliations:** 1Department of Public Health and Nursing, Norwegian University of Science and Technology, 7491 Trondheim, Norway; cecily.hu@outlook.com; 2Military Region 5 Hospital, Battambang, Cambodia; drsokhvannara@gmail.com (V.S.); sophy3366@gmail.com (S.N.); 3Trauma Care Foundation Cambodia, Battambang, Cambodia; yangvanheng@gmail.com; 4Tromsø Mine Victim Resource Center, University Hospital North Norway, 9038 Tromsø, Norway; husumhans@gmail.com (H.H.); roar.kloster@uit.no (R.K.); 5Department of Community Medicine, UiT the Arctic University of Norway, 9019 Tromsø, Norway; 6Department of Neurosurgery, University Hospital of North Norway, 9038 Tromsø, Norway; 7Center for International Health, Department of Global Public Health and Primary Care, University of Bergen, 5009 Bergen, Norway

**Keywords:** low-resource settings, neurosurgical capacity, traumatic intracranial hematoma, medical training, Cambodia

## Abstract

To evaluate the teaching effect of a trauma training program in emergency cranial neurosurgery in Cambodia on surgical outcomes for patients with traumatic brain injury (TBI). We analyzed the data of TBI patients who received emergency burr-hole trephination or craniotomy from a prospective, descriptive cohort study at the Military Region 5 Hospital between January 2015 and December 2016. TBI patients who underwent emergency cranial neurosurgery were primarily young men, with acute epidural hematoma (EDH) and acute subdural hematoma (SDH) as the most common diagnoses and with long transfer delay. The incidence of favorable outcomes three months after chronic intracranial hematoma, acute SDH, acute EDH, and acute intracerebral hematoma were 96.28%, 89.2%, 93%, and 97.1%, respectively. Severe traumatic brain injury was associated with long-term unfavorable outcomes (Glasgow Outcome Scale of 1–3) (OR = 23.9, 95% CI: 3.1–184.4). Surgical outcomes at 3 months appeared acceptable. This program in emergency cranial neurosurgery was successful in the study hospital, as evidenced by the fact that the relevant surgical capacity of the regional hospital increased from zero to an acceptable level.

## 1. Introduction

Cambodia is a Southeast Asian country with a total population of around 16.95 million in 2021 [1]. It is considered a lower-middle-income country, with a gross national income per capita in 2019 of 4320 USD [2]. According to the United Nations, 80% of Cambodians live in rural areas with a severe problem of generational poverty [1]. In 2015, the World Health Organization revealed that privately unqualified (nonmedical) providers dominate primary health care delivery in rural Cambodia at around 43% [3]. In recent years, the prevalence of traumatic brain injury (TBI) has increased in Cambodia [4], where the neurosurgical capacity is not up to international standard, with only 34 qualified neurosurgeons in the entire country in 2020 [5].

Most people in low- and middle-income countries, including Cambodia, are distributed in rural and remote areas with poor transportation, which result in extensive prehospital transit hours [6]. The long prehospital transport duration in turn leads to higher prehospital fatalities in rural settings [7,8]. As a solution to reduce trauma mortality, the University Hospital of Northern Norway (UNN) implemented medical training in remote areas where resources were scarce. Between 2002 and 2005, UNN initiated a training program in general trauma care in the Military Region 5 Hospital [9]. The Military Region 5 Hospital is a regional hospital in the most landmine-infested province in Cambodia; it acts as the main public referral trauma center for six provinces in the Northwest, with a catchments area of around 3 million people. Treatment at the hospital is open to all, and complete treatment for the poor population is granted free of charge.

After this training [9], two general surgeons from Military Region 5 Hospital began to carry out craniotomies on casualties with traumatic head injuries. To enhance the quality of the existing medical procedures in this regional hospital, UNN continued to provide training in emergency cranial neurosurgery, focusing on operative and postoperative care since September 2013. Box 1 describes the training in detail. The research team conducted a prospective, descriptive cohort study to examine the functional outcomes of patients treated by emergency trauma procedures at this hospital after the teaching intervention.

Box 1Teaching model in emergency cranial neurosurgery.
**Five 150-h training courses in primary trauma surgery**
Instructors from UNN conducted five consecutive 150 h training courses in primary trauma surgery at six regional hospitals in rural Cambodia from 2002 to 2005. The teaching program proved successful, with a significant reduction in trauma morbidity in the study hospitals [9]. The intervention was built on the Mozambique experience [10] and the previous experience of UNN’s “village university” teaching model [7,11,12].
**Cranial neurosurgical capacity-building program**
The training included a four-week in-hospital skill training at the neurosurgery department at UNN for a selected surgeon followed by teaching and surgical supervision for the surgeon’s team.In September 2013, senior surgeon Vannara Sokh from the Military Hospital underwent four weeks of in-hospital skill training at the neurosurgical department at UNN to enhance the quality of operative and postoperative care. Since October 2013, the hospital in Battambang had performed 14 craniotomies in TBI cases and 11 craniotomies in patients with cerebral hemorrhage. During this period, the neurosurgeon instructors at UNN guided the Cambodian surgical team online by evaluating CT scans and other patient information. They also made several visits to the study hospital, where they performed surgeries together with the local surgical team to provide hands-on teaching and supervision. The training program in emergency cranial neurosurgery did not introduce new techniques or procedures but controlled the quality of medical procedures already in use in the study hospital in Cambodia.

## 2. Materials and Methods

### 2.1. Eligibility Criteria for Intervention and Data Sources

All emergency procedures were performed by general surgeons Vannara Sokh (VS) and Sophy Nguon (SN) in the operating theatres of the study hospital. The study hospital had an intensive care unit (ICU) with basic equipment and drugs for postoperative care. The ICU had no ventilator service, and the researchers did not expect such service to be installed during the inclusion period. Therefore, patients with GCS < 7 were considered to have brain injuries that were too severe to profit from the craniotomy protocol applied at the study hospital.

The patients were included consecutively as they were admitted to the study hospital for emergency medical care after the injury/incident from January 2015 to December 2016. Computer tomography (CT) scans were performed on admission at a private clinic in Battambang. Photographs of the CT films were taken using a camera and sent to the clinical supervisors at UNN for evaluation. The Glasgow Coma Scale (GCS) was rated by one of the main investigators (VS or SN) on admission. The supervisors advised the surgical team in Cambodia and attended the decision to exclude cases from inclusion by two measures on admission: GCS < 7 and/or CT scans.

The endpoint was Glasgow Outcome Scale (GOS) three months after injury/incident. Study patients without full GOS ratings were excluded from the cohort study, resulting in anonymous data of 294 valid subjects being transferred to an Excel database for research analysis.

### 2.2. Data Collection

Baseline characteristics were gathered on case record forms at admission. Specific trauma data collection included overall anatomical severity (rated by injury severity score), brain injury severity (rated by GCS), and other medical factors. The severity of the brain injury was defined by GCS score as severe head injury (GCS ≤ 8), moderate head injury (GCS 9–12), or mild head injury (GCS 13–15) [13]. The injury severity score (ISS) is calculated to depict the severity of traumatic extracranial injuries. The higher the ISS score, the more severe the injuries. Patients with an ISS > 15 are considered severely injured [14]. The primary outcome was GOS three months after injury/incident, which was collected by the two investigators (VS and SN) through mobile telephone interviews or home visits to the study patients. The GOS is widely used to measure and monitor a patient’s recovery following head and nervous system injuries. A higher GOS score indicates better functional performance and independence (Table 1) [15].

### 2.3. Statistical Analysis

IBM SPSS Statistics for Windows version 26 was used for statistical analyses. Continuous variables were presented using means and standard deviations (SDs). Categorical data were presented using percentages and frequencies. We examined the data distribution using the Shapiro-Wilk test and the Q-Q plot. The Mann-Whitney U-test or the Kruskal Wallis test was applied to compare the differences between groups. For categorical variables, the chi-square test or Fisher’s exact test were used.

The GOS score as an outcome variable was subdivided into unfavorable (GOS 1–3) or favorable (GOS 4–5). The associations between the variables and the outcome were measured by univariate regression analysis. As we had a small sample size and a low number of outcomes, we could not perform a multivariable model with logistic regression. Complete case analysis was used for handling missing data. A significance level of *p* < 0.05 (two-tailed) was used for all analyses.

## 3. Results

### 3.1. Data Inclusion/Exclusion for Statistical Analysis

A total of 294 consecutive patients were enrolled in the cohort study for emergency burr-hole trephination or emergence craniotomy, including a few patients with GCS < 7. Based on our aim of focusing on trauma, nontraumatic admissions were excluded from the analysis, resulting in 235 valid subjects included in this analysis. A table comparing the subsets is available in the Appendix A. Figure 1 shows the process of data inclusion and exclusion for statistical analysis.

### 3.2. Patient Demographics and Clinical Status

The mean age at enrollment was 34 years, ranging from 12 to 84 years. Two hundred seven (88.1%) TBI patients underwent craniotomy. In the study, 83 (35.3%) patients presented a severe head injury (GCS ≤ 8), while 51 (21.7%) patients experienced a mild head injury (GCS 13–15). All patients had an ISS score of over 15.

In our study, 100 (42.5%) patients were diagnosed with an acute epidural hematoma (acute EDH), 74 (31.5%) with acute subdural hematoma (acute SDH), 35 (14.9%) with acute intracerebral hematoma (acute ICH), and 26 (11.1%) with chronic intracranial hematoma. Patients were predominantly males at 206 (87.7%), and this proportion ranged for the different trauma diagnoses from 82.4 to 94.3%. Patients with chronic intracranial hematoma were the eldest, with more than 85% of patients aged above 50 years. By contrast, patients with acute EDH were the youngest, with a mean age of 26 years. There was a difference in the time from injury to admission. The patient group with chronic intracranial hematoma experienced a delay between injury and admission, with an average duration from injury to admission of 484 ± 455 h, compared with an average duration of fewer than 60 h for each of the other trauma diagnoses. Before surgery, the patients with chronic intracranial hematoma had no polytrauma and almost no fractures. However, polytrauma was observed frequently in patients with acute intracranial hematoma (80%, 82.4%, and 74.3% for acute EDH, acute SDH, and acute ICH, respectively), and more than half of patients with acute intracranial hematoma had a fracture. Furthermore, all patients with acute EDH had fractures. No craniotomy was performed on patients with chronic intracranial hematoma, and almost no burr-hole trephination was performed on patients with acute intracranial hematoma (Table 2). Acute SDH and acute ICH mainly occurred after moderate or severe head trauma. In contrast, acute EDH mainly occurred after moderate head trauma, and chronic intracranial hematoma majorly occurred after mild head trauma (Table 2).

### 3.3. Outcome Measures

As shown in Table 3, a total of 218 (92.8%) patients experienced favorable outcomes. Favorable outcomes were achieved in 93 (93%) patients with acute EDH, compared with 66 (89.2%) with acute SDH, 34 (97.1%) with acute ICH, and 25 (96.2%) with chronic intracranial hematoma. All mortality rates reported were from the last follow-up, which was later than three months after injury, but the specific time was not recorded. All patients in a vegetative state (GOS = 2) on the fifth day after surgery and all patients with unfavorable outcomes at three months after the injury were deceased before the last follow-up (data not shown).

### 3.4. Risk Factors for Unfavorable Outcomes

As shown in Figure 2, the proportion of unfavorable outcomes (GOS = 1–3) at three months after injury was significantly higher in the group with a severe head injury than in the group with mild or moderate head injury (19.3% vs. 0% vs. 1%, *p* < 0.001). Univariate analysis using binary logistic regression showed that patients with moderate head injury were less likely to have an unfavorable outcome compared with those with severe head injury (OR 0.04, 95% CI 0.01–0.32). The preoperative GCS < 7 was associated with an unfavorable outcome at three months after injury (OR 26.3, 95% CI 7.9–87.1). In contrast, age, gender, type of fracture, trauma diagnosis, through referral hospital or not, surgery type, time from injury to admission, time from admission to surgery, and ISS score did not show significant associations with unfavorable outcomes (Table 4).

## 4. Discussion

In this study, we evaluated the surgical outcomes, patient characteristics, and prognosis predictors of emergency craniotomy and burr-hole trephination in a resource-constrained hospital in Cambodia.

### 4.1. Surgical Outcomes

The professionals at the hospital made the decisions regarding surgery type in the absence of clear guidelines for which surgical technique should proceed or which kind of patients should undergo neurosurgery. In the following sections, we compared the outcomes of surgical management by different trauma diagnoses with studies of high similarity (Table 5).

It is rare for epidural hematoma and intracerebral hematoma to become chronic [32,33,34]. Subdural hematoma is generated from the vein and has a slow and late onset, which is more likely to develop chronically [32]. Therefore, in the discussion section, the surgical outcomes of chronic intracranial hematoma in our study were compared with literature related to chronic subdural hematoma (CSDH).

Similar to other studies of CSDH patients [16,17,18,19,20], the surgical management for chronic intracranial hematoma in our study was burr-hole trephination. We observed a mortality rate of 3.8% in this patient group, which was notably lower than that in Singapore (in-hospital mortality rate of 13.3%) and in rural Kenya (30-day mortality rate of 6.7%) [16,19] but a bit higher than those in Taiwan (discharge mortality rate of 2.34%) and in China (mortality rate of 2% at six months after operation) [18,20]. In our study, the favorable outcome (GOS 4–5) was achieved in 96.2% of the patients with chronic intracranial hematoma, which was higher than that of surgically treated CSDH patients in Taiwan (83.3%) and in rural Kenya (90.8%) [16,20]. It is worth noting that more than half of CSDH cases were caused by head trauma [16,18,20]. However, most studies on the surgical management of CSDH included patients with injuries from all causes (both head injury and nontraumatic causes such as hypertension and alcohol consumption) [16,17,19,20]. It is interesting to note that surgical outcomes were similar in patients with nontraumatic CSDH and patients with traumatic CSDH [18]. In terms of mortality and favorable outcome, surgical outcomes in this study were acceptable compared with those in the published studies [16,17,18,19,20]. Therefore, we consider the surgical management of TBI patients with chronic intracranial hematoma at the Fifth Military Regional Hospital acceptable.

In our study, all patients with acute SDH underwent craniotomy as the surgical management. The postoperative mortality rate of 10.8% in the study hospital was very low compared with the mortality rate of 34.5% in Thailand (mortality at the last follow-up with a mean time of 6 months) and the 6-month mortality rate of 32–35.2% in the United Kingdom [21,22]. We observed a favorable outcome of 89.2% at three months, which seemed much better than 51% in Thailand and 42–45% in the United Kingdom [21,22]. It should be noted that the long-term results in our study were collected three months after the injury, while the long-term results of these two studies were collected six months after the injury [21,22]. The different time points of following up after the injury may explain the rate variations in favorable outcomes. Overall, we concluded that the surgical treatment of acute SDH patients at the Fifth Military Regional Hospital was of acceptable quality.

All acute EDH patients in our cohort underwent craniotomy except one patient who underwent burr-hole trephination. The surgical management in our study was similar to the published studies, which mainly involved craniotomy [24,26,27,28]. We observed mortality of 7%, which was slightly higher than the 2% across England and Wales [27] and the 3.3% in Hong Kong [24]. However, compared with studies in low-resource countries, the mortality rate in our study was lower than the 15.7% reported in Bangladesh [26] and similar to the 5.3% reported in Pakistan [28]. We found that the incidence of favorable outcomes three months after injury was 93%, which was much higher than the discharge outcomes in Japan (50%), Bangladesh (55.7%), and Hong Kong (76.7%) [24,25,26]. Therefore, we concluded that the surgical treatment of acute EDH patients in the Fifth Military Regional hospital was acceptable.

Almost all patients with acute ICH in our study received craniotomy. The practice was consistent with previous reports that craniotomy was the most used surgical method for traumatic intracerebral hematoma [29,30,35,36]. For surgically treated traumatic ICH patients, our favorable outcome of 97.1% was significantly higher than the 62% in the United Kingdom and the 63% reported in the first randomized trial (STITCH) for traumatic ICH patients in 31 centers from 13 countries [29,30]. Our mortality rate of 2.9% was prominently lower than the 10% in Brazil or the 15% reported in 31 centers from 13 countries [30,31]. Therefore, we concluded that the surgical treatment of traumatic ICH patients in the study hospital was acceptable.

### 4.2. Patient Characteristics: Mainly Young Males, High Incidence of Acute EDH, Late Admission

Our findings revealed a phenomenon that young men contribute the most to the cranial neurosurgery burden caused by TBI in Cambodia, which was consistent with the opinion of M. Kim et al. [5]. The unbalanced gender and age distribution of the primary labor force, the high proportion of men among drivers, and the high ratio of men who engage in violence or fighting could explain the phenomena observed in our study patients [1,37,38,39].

Acute EDH was the most frequent diagnosis in our study (43%). Many authors reported that the patient population with acute EDH was the youngest among all trauma diagnoses [26,40,41]. The high incidence of acute EDH may be related to the demographic characteristic that more than half of Cambodia’s population was under the age of 22 [1]. Kim et al. in 2020 [5] also documented that EDH accounted for a slightly higher proportion than SDH in TBI patients admitted to a public hospital in Cambodia.

Late admissions were observed in our study, which may be due to the absence of either an ambulance system or a referral system in Cambodia. Consequently, many patients were admitted to emergency surgery many days and even weeks after the injury [5]. Delays may also be due to resource constraints, such as lack of appropriate beds in receiving hospitals, lack of vehicles, or lack of trained referral staff [42,43].

### 4.3. Risk Factors of Unfavorable Outcomes

Kulesza et al. and Hanif et al. suggested that age and preoperative GCS score could predict the outcome after TBI, but gender did not have predictive value [44,45]. In our study, the severity of head injury based on GCS score was the only factor that was associated with the surgical outcomes (long-term unfavorable outcomes). Age was repeatedly identified as a predictor of TBI prognosis, and older age was associated with unfavorable GOS outcomes [14,44,46]. Hanif et al. commented that the likelihood of poor outcomes following surgical intervention in older patients with TBI was even magnified [45]. They further concluded that prognosis was also associated with time from injury to treatment and surgery type [45]. Researchers have noted that for traumatic acute intracranial hematoma requiring surgical evacuation, the more expediently (less transfer time) the surgical intervention is performed, the better the outcome [47,48]. However, we did not find that factors such as age, time from injury to surgery (injury to admission, admission to surgery), or surgery type played a role in predicting postoperative outcome.

### 4.4. Strengths and Limitations

The reasons for the lower mortality rates we observed compared with those reported in other studies may relate to the fact that study patients who lost contact after discharge were not included in the data set (due to lack of endpoint data). Cambodian society is disorganized, with many migrant families, which may have influenced the dropout rate. Moreover, the patients in Cambodia may be generally younger, and the younger age group may carry a better prognosis. Furthermore, a greater proportion of severe patients were not enrolled in the intervention due to resource constraints in the study hospital.

To our knowledge, this is the first study to evaluate the teaching outcome of a neurosurgical capacity-building program in Cambodia. Two Cambodian investigators managed all the procedures in the study hospital. As such, we did not expect disagreements and variations between raters and performers in the study hospital. Additionally, ours was a prospective study design, which ensured the accuracy of data collection. There were several limitations related to our study. One was that all cranial cases were CT scanned outside the hospital due to the lack of CT capability at the study site. Therefore, underdiagnosis bias may exist. Another was that this was a single-center study, which may reflect only the experience of the study hospital, resulting in limited generalizability. The other limitation was that the outcome measurements were unblinded and performed by the surgeons themselves, which may have led to misclassified results and biased assessments. Despite these limitations, our findings demonstrate a complex treatment reality that could serve as a framework for future training programs.

## 5. Conclusions

We found that the emergency burr-hole trephinations and emergency craniotomies for TBI patients at the Fifth Military Regional Hospital resulted in acceptable outcomes and that severe traumatic brain injury was identified as a predictor of the study hospital’s long-term unfavorable outcomes.

## Figures and Tables

**Figure 1 ijerph-19-06471-f001:**
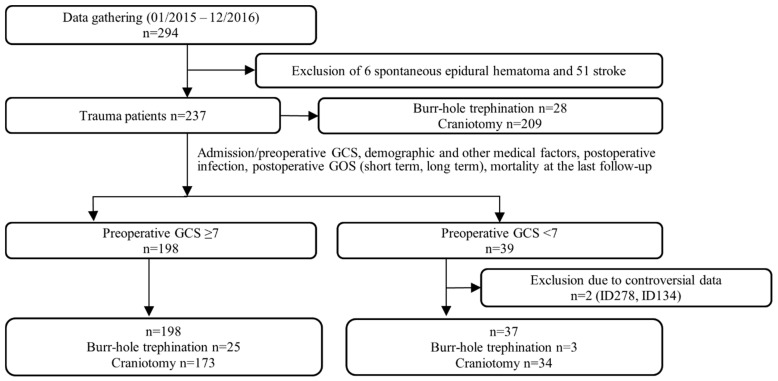
Flowchart of the process of data inclusion and exclusion for statistical analysis. Overview of the process of extracting data for the study patient of interest. Abbreviations: GCS, Glasgow Coma Scale; GOS, Glasgow Outcome Scale; ISS, Injury Severity Score.

**Figure 2 ijerph-19-06471-f002:**
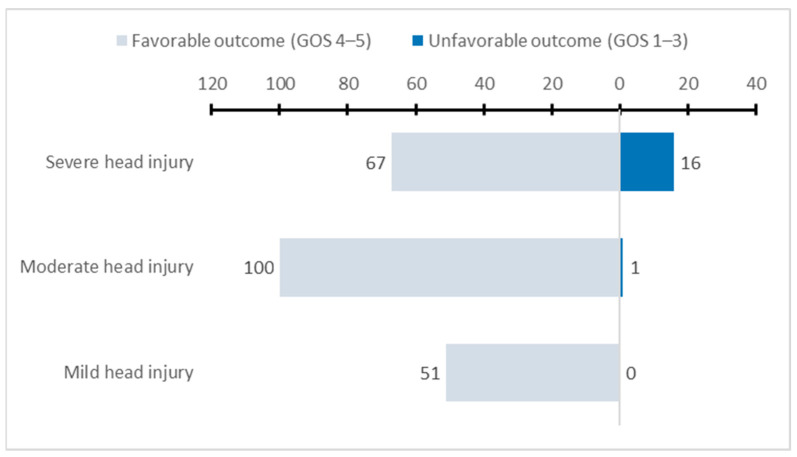
The number of patients with a favorable outcome at three months after injury in the different preoperative GCS groups.

**Table 1 ijerph-19-06471-t001:** The Glasgow Outcome Scale.

Rating	Interpretation
1	Dead
2	No response, but alive
3	Severe disability, conscious but needs support for all daily activities
4	Patient is independent but has some disability
5	Full recovery, no disability

**Table 2 ijerph-19-06471-t002:** Baseline characteristics of the study population grouped by trauma diagnosis.

	Overall	Trauma Diagnosis				
Characteristic	N = 235	Epidural Hematoma	Subdural Hematoma	Intracerebral Hematoma	Chronic	*p*-Value
		*n* = 100	*n* = 74	*n* = 35	*n* = 26	
Age	34.9 ± 17.3	26.3 ± 11.3	37.2 ± 16.8	36.2 ± 15.3	59.7 ± 13.7	<0.001 ^a^
Time from injury to admission (h)	83 ± 213.5	25.3 ± 33.3	37.1 ± 99	58.5 ± 91.3	484.2 ± 455.6	<0.001 ^a^
Time from admission to surgery (h)	21.4 ± 36.4	17.2 ± 25.1	19.3 ± 28	24.5 ± 29.6	39.3 ± 77.5	0.7 ^a^
Injury Severity Score	23.2 ± 3.6	23.5 ± 3.4	23.5 ± 3.3	22.7 ± 4	22.2 ± 4.2	0.279 ^a^
**Head injury severity**						<0.001 ^b^
Severe (GCS 3–8)	83 (35.3%)	29 (29%)	33 (44.6%)	15 (42.9%)	6 (23.1%)	
Moderate (GCS 9–12)	101 (43.0%)	50 (50%)	33 (44.6%)	12 (32.3%)	6 (23.1%)	
Mild (GCS 13–15)	51 (21.7%)	21 (21%)	8 (10.8%)	8 (22.9%)	14 (53.8%)	
**Gender**						0.353 ^c^
Female	29 (12.3%)	11 (11%)	13 (17.6%)	2 (5.7%)	3 (11.5%)	
Male	206 (87.7%)	89 (89%)	61 (82.4%)	33 (94.3%)	23 (88.5%)	
**Age groups**						<0.001 ^c^
<20 y	41 (17.4%)	31 (31%)	5 (6.8%)	5 (14.3%)	0 (0%)	
20–34 y	105 (44.7%)	53 (53%)	38 (51.4%)	12 (34.3%)	2 (7.7%)	
35–49 y	29 (12.3%)	8 (8%)	9 (12.2%)	10 (28.6%)	2 (7.7%)	
50–64 y	40 (17%)	7 (7%)	15 (20.3%)	7 (20%)	11 (42.3%)	
> or =65 y	20 (8.5%)	1 (1%)	7 (9.5%)	1 (2.9%)	11 (42.3%)	
**Type of fracture**						<0.001 ^c^
Close fracture	130 (55.3%)	98 (98%)	16 (21.6%)	15 (42.9%)	1 (3.8%)	
Open fracture	8 (3.4%)	2 (2%)	3 (4.1%)	3 (8.6%)	0 (0%)	
Without fracture	97 (41.3%)	0 (0%)	55 (74.3%)	17 (48.6%)	25 (96.2%)	
**Polytrauma**						<0.001 ^b^
No other injuries	68 (28.9%)	20 (20%)	13 (17.6%)	9 (25.7%)	26 (100%)	
Other moderate injury no need for surgery	162 (68.9%)	77 (77%)	60 (81.1%)	25 (71.4%)	0 (0%)	
Other injury in need for surgery	5 (2.1%)	3 (3%)	1 (1.4%)	1 (2.9%)	0 (0%)	
**Type of operation**						<0.001 ^c^
Burr-hole trephination	28 (11.9%)	1 (1%)	0 (0%)	1 (2.9%)	26 (100%)	
Craniotomy	207 (88.1%)	99 (99%)	74 (100%)	34 (97.1%)	0 (0%)	

Abbreviations: GCS, Glasgow Coma Scale. ^a^. Kruskal Wallis test; ^b^. Chi-square test; ^c^. Fisher’s exact test if more than 20% of cells have an expected count of less than 5 on the chi-square test.

**Table 3 ijerph-19-06471-t003:** Trauma Diagnoses and Outcome Measures.

	Overall	Trauma Diagnosis				
Characteristic	N = 235	Epidural Hematoma	Subdural Hematoma	Intracerebral Hematoma	Chronic	*p*-Value ^a^
		*n* = 100	*n* = 74	*n* = 35	*n* = 26	
**Dead**						0.540
Yes	17 (7.2%)	7 (7%)	8 (10.8%)	1 (2.9%)	1 (3.8%)	
No	218 (92.8%)	93 (93%)	66 (89.2%)	34 (97.1%)	25 (96.2%)	
**Postoperative GOS (3 months post-injury)**						0.540
Unfavorable (GOS 1–3)	17 (7.2%)	7 (7%)	8 (10.8%)	1 (2.9%)	1 (3.8%)	
Favorable (GOS 4–5)	218 (92.8%)	93 (93%)	66 (89.2%)	34 (97.1%)	25 (96.2%)	

Abbreviations: GOS, Glasgow Outcome Scale. ^a^. Fisher’s exact test if more than 20% cells have expected count less than 5 on the chi-square test.

**Table 4 ijerph-19-06471-t004:** The risk of unfavorable outcome three months after injury: univariate analysis by binary logistic regression.

	Unfavorable Outcome (GOS 1–3) at 3 Months after Injury			
Variable	*p*-Value	OR	95% CI for OR	
			Lower	Upper
Moderate head injury (GCS 9–12)	-	ref	-	-
Severe head injury (GCS 3–8)	**0.002**	23.88	3.09	184.36
Mild head injury (GCS 13–15)	0.998	0 ^a^	0	-
Female	-	ref	-	-
Male	0.156	0.42	0.13	1.39
Type of fracture (Close fracture)	-	ref	-	-
Type of fracture (Open fracture)	0.76	1.41	0.16	12.4
Type of fracture (No fracture)	0.147	0.42	0.13	1.35
Trauma diagnosis (Epidural hematoma)	-	-	-	-
Trauma diagnosis (Subdural hematoma)	0.379	1.61	0.56	4.66
Trauma diagnosis (Intracerebral hematoma)	0.388	0.39	0.05	3.29
Trauma diagnosis (Chronic)	0.563	0.53	0.06	4.52
Referral admission (No)	-	ref	-	-
Referral admission (Yes)	0.849	1.11	0.39	3.1
Polytrauma (No)	-	ref	-	-
Polytrauma (Yes)	0.295	1.98	0.55	7.13
Age group (<20 y)	-	ref	-	-
Age group (20–34 y)	0.177	4.21	0.52	33.99
Age group (35–49 y)	0.385	2.96	0.26	34.32
Age group (> or =50 y)	0.356	2.86	0.31	26.53
Surgery type (Burr-hole trephination)	-	ref	-	-
Surgery type (Craniotomy)	0.437	2.26	0.29	17.75
ISS groups (ISS < 25)	-	ref	-	-
ISS groups (ISS ≥ 25)	0.172	4.16	0.54	32.22
Time from injury to admission (Hours)	0.589	1	1	1
Time from admission to surgery (Hours)	0.453	0.99	0.97	1.02

Abbreviations: GCS, Glasgow Coma Scale; GOS, Glasgow Outcome Scale; ISS, Injury Severity Score; OR, Odds ratio; CI, Confidence interval; Findings with *p*-value ≤ 0.1 are highlighted in bold; ^a^ For mild head injury, there was no case suffered unfavorable outcome.

**Table 5 ijerph-19-06471-t005:** Summary of the literature comparisons of the surgical outcomes.

Trauma Diagnosis	Surgical Outcomes in the Study Hospital	Surgical Outcomes * in the Literature on the General Population	Surgical Outcomes * in the Literature on a Specific Population
Chronic subdural hematoma	Favorable outcome at 3 months 96.2% Mortality 3.8%	Favorable outcome 90.8% [16] Mortality 0–32% [16,17,18,19]	Age ≥ 65 years [20]: Favorable outcome 83.3% Mortality 2.34%
Traumatic acute subdural hematoma	Favorable outcome at 3 months 89.2% Mortality 10.8%	Favorable outcome 42–51% [21,22] Mortality 32–35.2% [21,22]	Comatose patients (GCS < 10) [23]: Favorable outcome 23% Mortality 57%
Traumatic acute epidural hematoma	Favorable outcome at 3 months 93% Mortality 7%	Favorable outcome 50–76.7% [24,25,26] Mortality 2–15.7% [24,26,27,28]	Not available
Traumatic intracerebral hematoma	Favorable outcome at 3 months 97.1% Mortality 2.9%	Favorable outcome 62–63% [29,30] Mortality 10–15% [30,31]	Not available

* Surgical results at discharge, or surgical results at follow-up within 30 days after surgery, or 3 months after trauma, or at 6 months after trauma.

## Data Availability

The datasets generated and/or analyzed during the current study are not publicly available due to data privacy but are available from the corresponding author upon reasonable request.

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
