# Peer review of "Emergency Craniotomy and Burr-Hole Trephination in a Low-Resource Setting: Capacity Building at a Regional Hospital in Cambodia"

_ijerph, 2022, doi:10.3390/ijerph19116471_

Round 1

Reviewer 1 Report

The Authors carrtied out an extensive revision of their manuscript, in paritcular regarding  the statistical aspects and resuslts. The conclusions are sound and relevant  especially  regarding the neurosurgical training  of medical staff and the organization of neurosurgical assistance to the severely head inijured people in the developing countries.

Reviewer 2 Report

Thank you for considering previous remarks and revising your manuscript which has improved considerably. However, please find below some comments:

1/ In the abstract, please mention that the outcome was the GOS

2/ I would remove “Interventional” from the abstract and the study as it was an observational study and you did not measure the effect of the “intervention (surgery)” by comparing it to standard care.

3/ You cannot say that GCS<7 was an exclusion criteria and include those patients (or a part of them) in the study, even if you mention that it was a protocol violation. All the more so as GCS<7 was associated with unfavorable outcome. It is very confusing. You only can specify that NN patients with GCS<7 were excluded from the study because they “were considered having a brain injury too severe to profit from the craniotomy protocol applied at the study hospital” and add this point as a limit.

Thus, please remove this sentence: “As a protocol deviation in the cohort study, a few patients with clinical signs of severe brain injury, GCS < 7, were included in the intervention. In our branch of research (for the trauma population), we tried to exclude as few study units as possible, since there is in-formation to learn from. Thus, the subset of TBI patients with preoperative GCS < 7 was not excluded, resulting in 235 valid subjects included in this analysis. A table comparing the subsets is available in the Supplementary Materials.”

4/ “Study patients without full GOS rating were excluded from study” : the number of those patients excluded because the outcome was unavailable should be mentioned in the flowchart and this should also figure in the limits section

5/ The Flowchart should be redone and placed only in the results section. It should mention at the top the number of eligible patients, then the number of excluded patients at each step should be mentioned (too severe, CT findings, missing or controversial data…). After all exclusion criteria, you have your 235 included subjects and you can split them into GCS< or >7 …

6/ “Demographic and medical data were gathered on Case Record Forms at admission, before surgery, 1-5 days after surgery, and three months after injury/incident (see Supplementary Form S1)”: I didn’t see any post operative data (days 1-5…) in the results section except for the outcome. Please change by: “Baseline characteristics were gathered on Case Record Forms at admission”

7/ Line 146: The number 207 should be written in letters as it starts a sentence.

8/ Line 180: Please remove from the result section (it should be in the discussion only): “It is rare for epidural hematoma and intracerebral hematoma to become chronic [16-18]. Subdural hematoma is generated from the vein and has a slow and late-onset, which is more likely to develop chronically [16]. Therefore, in the discussion section, the surgical outcomes of chronic intracranial hematoma in our study were compared with literature 183 related to chronic subdural hematoma (CSDH).”

9/ Line 193: there is a mistake: You wrote “unfavorable outcomes (GOS = 1-2)” Istead of ”(GOS 1-3)”

10/ Finally, you chose not to report the multivariable logistic regression. Thus, you have to remove this sentence from the analysis section: « The independent risk factor was identified using binary logistic regression. The potential confounding variables were adjusted using a multivariable regression model. They were recognized at the p-value cut-off point of 0.1 in the univariate regression analysis or based on previous literature. To minimize small sample bias, at least 10 events are required per predictor (EPV) in the regression model [15]. Multivariable regression model Complete case analysis was used for handling missing data. » And replace it by a sentence like : “Association between variables and the outcome was measured or assessed by univariate regression analysis. As we had a small sample size and a low number of outcomes, we couldn’t perform a multivariable model with logistic regression” And remove the last sentence (line 203-206) from the results.

11/ Line 307-323 is duplicated

12/ In the limits you should add that the fact that outcome measurements was unblinded and performed by the surgeons itself could result in misclassified outcomes and biased measure.

Thank you

Reviewer 3 Report

I appreciate the attempted revision. My concern still stands. No surviving patients had a bad outcome? This is not consistent with severe TBI.

Author Response

This manuscript is a resubmission of an earlier submission. The following is a list of the peer review reports and author responses from that submission.

Round 1

Reviewer 1 Report

The Authors did a good and interesting job. They report the eperience at Military regional 5 Hospital - without CT facility - about neurosurgical treatment of 294 cases of TBI, performed by  general surgeons who have recived a specific training.Overall" acceptable" results are reported.This  aspect should be correlated with the statement :"a greater proportion of severe patients may not undergo surgery.

Reviewer 2 Report

Review :

Thank you for the opportunity to review this manuscript evaluating the outcomes of patients with traumatic brain injury (TBI) attending a general trauma care in a regional hospital in Cambodia which is located in a remoted area not benefiting from neurosurgical capacity up to international standards. In order to transfer knowledge, the university hospital implemented a medical training program to teach neurosurgical procedures to regional hospitals so that they can take care of patients with TBI.

INTRODUCTION

The aim of the study is not totally clear for me. At the end of the introduction, authors state that « The major aim of this cohort was to systematically evaluate the outcome of the teaching intervention by the rating of functional outcome in the study patients, and expert evaluation of the technical performance during the craniotomies. In this study, we aimed to systematically evaluate the outcome of the teaching intervention by the rating of functional outcome in the studied patients”

I do not think you can evaluate the teaching intervention because there is no comparator (for example before and after intervention or patients attended by trained and not trained physicians). In this way, the study is not “interventional” but only descriptive, even if it is prospective. Thus, I suggest to authors that they amend this statement and say that:

“The research team conducted a prospective, descriptive cohort study to examine the functional outcome of patients treated by emergency trauma procedures at this hospital after the teaching intervention.”

Instead of :

« The research team conducted a prospective, descriptive interventional cohort study (Jan 2015 to Dec 2016) to examine the outcome of the teaching intervention of emergency craniotomies at this hospital. The major aim of this cohort was to systematically evaluate the outcome of the teaching intervention by the rating of functional outcome in the study patients, and expert evaluation of the technical performance during the craniotomies. In this study, we aimed to systematically evaluate the outcome of the teaching intervention by the rating of functional outcome in the studied patients.”

And to report the study period “Jan 2015 to Dec 2016)” in the method section.

I suggest to the authors to remove Figure 1.

METHODS

In the Method section, the inclusion criteria should be described (age? Eligible criteria for an emergency procedure?...). Thus, authors should describe briefly the journey of patient with TBI in their hospital. Who were eligible for craniectomy? Was craniectomy feasible 24/7 (on which delay were the 2 surgeons that performed all the procedures available). Were the procedures were performed? It would be interesting to report the prognosis of patients with TBI that didn’t have craniectomy.

Therefore, the flowchart should mention, all the patients that were attended for TBI, and, among them, those who were eligible for an emergency procedure, and, among them, those who had an emergency procedure and were included in the study. It is important for the external validity to know to how many eligible patients, these emergency procedures apply. And maybe the background and experience in neurosurgery of the 2 physicians that performed the procedures.

The number of included patients and the flowchart should be in the result section and not in the methods.

Please, note “Glasgow Coma Scale (GCS)” when abbreviation is cited at the first time (Data collection) in order to reduce the risk of confusion with GOS.

As the GOS is the primary outcome, authors should describe it for authors that are not familiar with it (maybe in a table?). And how this GOS was assessed? By who?

A severe head injury was defined by a GCS ≤ 8 whereas the cut-off is 7 in the flow-chart. Why didn’t you choose the same cut-offs?

Maybe replace “Injury/incident” by “Injury”

In the logistic regression, authors stated that “time from injury” and “age” were introduced in the model as confounders. But which variable was the explanatory variable of interest? Were there several explanatory variables? How did authors choose those variables in addition to the confounders cited above? How they remove or add variables to obtain their final model? When reading the results, I understand that the only explanatory variable was pre-operative GCS and that you performed 2 models, one adjusting for age and another adjusting for time from injury to admission. Why 2 models instead of 1 with the 2 confounders. This should be explained here.

As complete case analysis was used, it should be interesting to report how many patients were removed from the final model due to missing data.

RESULTS

This section should begin by the number of included patients and the flowchart.

Please remove decimals from years in age.

Please give N and %, not just only %.

The % addition of the four types of trauma diagnoses is 100,1%

I suggest to simplify the sentence: “The enrolled patients were not only with a very marked male predominance in general (87.7%) but were also predominantly male among these four types of trauma diagnoses (male proportions in the range of 82.4-94.3%, respectively).” By: “Patients were predominantly males NN (87.7%) and this proportion ranged in the different types of trauma diagnoses from 82.4 to 94.3%, respectively”

I suggest removing “considerable” or “extreme” when reporting delays. Those subjective adjectives should be used in the discussion rather than in the results.

Replace “incident” by “injury”.

Please remove the decimals from 484.2 ± 455.6 hours.

Please could you define “mildest condition” or remove it. The comparison of patients with or without polytrauma or fracture is sufficient and self-explanatory.

Please could you replace the Figure 3 by bar plots? Or maybe add this information in Table 2. Therefore, the information is already in the table (pre-operative GCS) but with other cut-offs. For simplification, I recommend to remove the figure and use in the Table 2 the same cut-offs that were defined in the method section for injury severity (and that are reported in the figure). If authors prefer to show a figure, the replace the pie chart by barplots and remove pre-operative GCS from the table to avoid redundant information.

Please give the total number (and%) of patients with unfavorable/favorable outcomes and also give the NN (%%) in each type of trauma diagnosis group.

I suggest changing “The proportion of favorable outcome (GOS = 4-5) at three months after injury/inci dent (98% vs. 64.9%, p < 0.001) was significantly higher in the group with preoperative GCS ≥ 7 compared to that in the group with preoperative GCS < 7 (Figure 4)” by “As shown in Figure 4, the proportion of favorable outcome (GOS = 4-5) at three months after injury was significantly higher in the group with preoperative GCS ≥ 7 compared to that in the group with preoperative GCS < 7 (98% vs. 64.9%, p < 0.001)”

I suggest to let only 1 decimal in ORs (for ex 26.3 instead of 26.27).

I suggest reapproaching the first sentence with the sentence giving the OR for GCS:

“As shown in Figure 4, the proportion of favorable outcome (GOS = 4-5) at three months after injury was significantly higher in the group with preoperative GCS ≥ 7 compared to that in the group with preoperative GCS < 7 (98% vs. 64.9%, p < 0.001). Univariate analysis using binary logistic regression showed that only the preoperative GCS < 7 was associated with an unfavorable outcome at three months after injury (OR 26.3, 95% CI 7.9–87.1). In contrast, age, gender, type of fracture, trauma diagnosis, through referral hospital or not, surgery type, time from injury to admission, time from admission to surgery, and ISS score were not.”

Why did authors perform 2 models instead of one with the 2 confounding variables?

I didn’t see any data on mortality in the results (only in the discussion). I also didn’t see either any data on adverse effects of those emergency surgical procedures.

DISCUSSION:

I suggest to shorten the first part “mainly young males…”.

Abbreviation CSDH should be used earlier in the methods/results.

In the table 3, did all the patients undergo a neurosurgical procedure? It would be of interest to give the prognosis (fatality rate, GOS) of patients with severe TBI that do not undergo such procedures.

I would have put the discussion comparing outcomes (mortality and GOS) with other study sooner in the discussion. Data on study mortality should be given in the results section.

In risk factors, authors stated that GCS score was associated to poor surgical outcomes including “postoperative wound infection” but I didn’t see this outcome and association in the results

Maybe authors didn’t show an association between age and outcome because the patients included in this study were very young. Were patients older in the study of Hanif et al?

CONCLUSION:

I suggest shortening the conclusion to the 2 first sentences.

Reviewer 3 Report

The authors report on patients with traumatic intracranial pathology in Cambodia over a two-year period. I have some minor and some major concerns. It's an interesting subject, but the analyses are problematic.

Major:

  1. The authors appear to have cherry-picked the threshold of GCS7 for binary yes/no in their regressions. It would be better to use severe vs. moderate vs. mild.
  2. The suggestion is that these two surgeons have the best outcomes in the history of these procedures. 90% good outcomes at 3 months after acute subdural hematoma, 97% (!) after traumatic ICH is difficult to believe. To the point that the authors would need to be more granular with the data (hematoma volumes, post-op scans, pupil exams, GCS motor exams, discussion of ICP monitoring, etc) to begin to be believable. 

Minor:

  1. More details on referrals is required. How to patients get sent to their hospital? Do all patients have CT scans prior to transfer? Do all patients have CT scans period? How many patients died in transit?
  2. More details on outcomes assessments is required. When are these patients transferred back home? How are they seen for follow-up outcomes assessment?
  3. Box 1. How many hours of the training courses are devoted to neurosurgery?
  4. Mechanism of injury is important. It should be captured.
  5. I really don't think it's appropriate to lump chronic subdurals in with these acute pathologies. It's not interesting to talk about delay in intervention for a chronic subdural, it's delayed by definition (it's chronic).